# Photogrammetric Method to Determine Physical Aperture and Roughness of a Rock Fracture

**DOI:** 10.3390/s22114165

**Published:** 2022-05-30

**Authors:** Masoud Torkan, Mateusz Janiszewski, Lauri Uotinen, Alireza Baghbanan, Mikael Rinne

**Affiliations:** 1Department of Civil Engineering, Aalto University, 02150 Espoo, Finland; mateusz.janiszewski@aalto.fi (M.J.); lauri.uotinen@aalto.fi (L.U.); mikael.rinne@aalto.fi (M.R.); 2Department of Mining Engineering, Isfahan University of Technology, Isfahan 8415683111, Iran; bagh110@cc.iut.ac.ir

**Keywords:** photogrammetry, physical aperture, roughness, JRC, camera, rock fracture

## Abstract

Rock discontinuities play an important role in the behavior of rock masses and have a high impact on their mechanical and hydrological properties, such as strength and permeability. The surfaces roughness and physical aperture of rock joints are vital characteristics in joint shear strength and fluid flow properties. This study presents a method to digitally measure the physical aperture of a rock fracture digitized using photogrammetry. A 50 cm × 50 cm rock sample of Kuru grey granite with a thoroughgoing fracture was digitized. The data was collected using a high-resolution digital camera and four low-cost cameras. The aperture and surface roughness were measured, and the influence of the camera type and 3D model rasterization on the measurement results was quantified. The results showed that low-cost cameras and smartphones can be used for generating 3D models for accurate measurement of physical aperture and roughness of rock fractures. However, the selection of appropriate rasterization grid interval plays a key role in accurate estimations. For measuring the physical aperture from the photogrammetric 3D models, reducing rasterization grid interval results in less scattered measurement results and a small rasterization grid interval of 0.1 mm is recommended. For roughness measurements, increasing the grid interval results in smaller measurement errors, and therefore a larger rasterization grid interval of 0.5 mm is recommended for high-resolution smartphones and 1 mm for other low-cost cameras.

## 1. Introduction

The geometrical properties of rock fractures such as aperture, roughness, orientation, length, persistence, and spacing are important in rock engineering projects since they have a great impact on rock mass mechanical and hydrological properties like strength and permeability. The surfaces roughness and physical aperture of rock fractures are key characteristics in fracture shear strength and fluid flow investigation. Rock mass hydraulic and shear strength can be affected by roughness, especially in clean and unfilled fractures [1]. As a result, determining the physical aperture and roughness of rock fractures is critical for determining the engineering stability of rock masses.

The joint roughness coefficient (JRC) is a widely used parameter representing the degree of roughness introduced by Barton [2]. The JRC is usually back-calculated using a tilt or shear test. These tests are frequently time-consuming and costly. To address this problem, a set of 10 standard roughness profiles was proposed by Barton and Choubey, representing different JRC values ranging from 0 to 20 [1]. These profiles were approved as a standard method to determine fracture roughness by the International Society of Rock [3], in which visual comparison with these profiles is used to estimate a JRC value, which is rather subjective [4,5,6]. Numerous studies have tackled this subjective evaluation for gaining exact JRC values. This led to many empirical equations based on different approaches, e.g., fractal dimension and other methods, which have been summarized in refs. [7,8].

The most used method is the Z_2_, where the root mean square of the first deviation of the profile is computed [9,10,11,12,13,14,15,16,17]. The second method is the roughness profile index (Rp), which is calculated as the ratio of the real length of fracture profile to the projected length in the fracture plane [4,10,12]. The third approach is the structure function (SF), which quantifies the variation in surface texture [4,12,14]. Another approach uses the fractal dimension (D) as a degree of surface variation [12,18,19]. The fifth approach (θ*_max_/(C + 1)_2D_ computes the maximum inclination (θ*_max_) and a dimensionless parameter C calculated from the relationship between the normalized length and the inclination threshold [12,20]. The parameters tend to be very sensitive to the sampling interval [10].

The original JRC profiles have been digitized at different interval ranges by employing different techniques, for instance, digitizer [4,9], image processing [13,15,16], AutoCAD [20], Origin software [12], and GETDATA software [17]. As a result, further research on fracture topography shifts away from visual comparison with standard 2D profiles toward measurement of 3D surfaces and internal fracture geometry using various approaches.

Early investigations employed destructive methods to measure the internal fracture geometry such as injection of epoxy into the fracture and slicing the sample (e.g., [21]). A range of non-destructive methods was also applied, such as X-ray computed tomography scanning [22,23], magnetic resonance imaging [24,25], or transmitted light method for transparent fracture analogs [26,27]. Later, high-resolution, precise scanning devices were utilized to measure the surface topography and aperture of rock fractures in the laboratory, such as optical profilometers (e.g., [28]), 2D microscope cameras (e.g., [29]), or 3D laser scanners (e.g., [29,30]).

Surface properties of rock fractures can also be measured from digitized rock surfaces created using Structure-from-motion (SfM) photogrammetry [31,32,33,34,35,36,37,38,39,40,41,42,43,44,45,46,47,48,49,50,51,52,53,54,55,56]. For instance, SfM enables estimation of the surface roughness [31,32], and the effect of the type of cameras, sensors, and lenses on roughness measurements was investigated in refs. [33,42,51,53,54,55]. The SfM method allows investigation of the reliability of JRC values with high-resolution images [34,35,36,37,51], and photogrammetric JRC estimation for the design and stability analysis of slopes [43,44]. The SfM is advantageous as it allows estimation of JRC error models for both laboratory and field photogrammetric surveys [41,42,46]. Other applications of SfM related to rock fracture characterization are prediction of peak shear strength [47], using 3D models obtained by photogrammetry for simulation of fluid flow through a fracture [48], and validation of photogrammetry data with pull and push shear tests to predict friction angle [45,50]. The method was applied successfully for different materials [51,52,53], different sample size [53], and the quality of the reconstructed model was investigated by the angular distance between subsequent images [56]. This efficient and low-cost remote sensing technique generates high-resolution digital 3D models of rock surfaces from a set of images. As a result, accurate measurements can be derived from the 3D models [49]. Nonetheless, the trustworthiness of aperture and roughness measurements from 3D models are governed by the quality of the images, and of the photogrammetric reconstruction process [34].

However, such techniques for obtaining 3D representations of rock fracture surfaces require expensive equipment with a limited scanning area, which limits their widespread use in rock fracture surface measurement of large-scale samples. As a result, there is a great demand for a low-cost method of obtaining 3D data of rock fracture surface properties. This is particularly important when low-cost cameras are used to collect the data. However, only a limited number of studies exist that quantify the effect of low-cost photogrammetric process on fracture measurements, e.g., [51]. Nowadays, smartphones’ cameras and processors are being developed rapidly. This opens the possibility for many people to have access to low-cost and high-quality cameras. In addition, several photogrammetric software packages exist for reconstructing 3D models, and their processing speed has increased considerably in recent years. The packages provide a great opportunity for researchers to do their research with more accessible tools. In addition, no comprehensive study has been found in the literature that describes a method for the measurement of physical aperture of rock fracture from the photogrammetric models. This can be attributed to the difficulty of the process, as it requires accurate transformation and matching of the top and bottom half of the digitized sample.

In this study, an attempt has been made to provide a methodology to characterize physical aperture using photogrammetry as a non-destructive method. Besides, a comparison study was conducted to identify the effect of using different photography devices with different resolutions and costs on characterizing geometrical properties of natural rock fractures. Using markers gives the opportunity to put two halves of a rock sample in the same coordinate system to calculate physical aperture and improve the accuracy of 3D models. In addition, the effect of the rasterization grid intervals of the rough surfaces was investigated on an estimation of physical aperture and roughness value. A proper rasterization grid interval could help in order to obtain more precise results and better quality of the point clouds.

The paper presents a workflow to measure physical aperture and estimate the roughness using different photographic tools. The comparison was done between the results obtained by a DSLR and low-cost cameras of a rock fracture in a 50 cm × 50 cm sample. The physical aperture is calculated using the cloud-to-cloud distance of the top and bottom models and the surface roughness is calculated using the Z_2_ approach [9] and JRC formulas proposed by Yu and Vayssade [10]. The effects of the photogrammetric reconstruction and model rasterization on the measured aperture and roughness are assessed. Finally, some of the benefits and drawbacks of employing this method are highlighted.

## 2. Materials and Methods

The rock fracture is digitized using photogrammetry and a rotating table with five cameras. The 3D model of the rock sample is reconstructed from the images and analyzed to compute the aperture and roughness of the fracture surface. The workflow overview is given in Figure 1. To conduct hydromechanical or shear tests, rock is frequently split into two pieces with a single artificial fracture (Figure 1a). Dimensions of the rock were selected based on some limits, such as experimental methods or facilities. The distance between two pieces is called physical aperture and it plays a key role in the abovementioned tests. The photogrammetry was conducted as follows: the whole sample, when two halves are tightened together (Figure 1d), top and bottom halves separately (Figure 1e). As a result, there were three point clouds, the top half, the bottom half, and the whole sample. Distances between markers were specified in the 3D model to scale the whole sample (Figure 1f) and a coordinate system was defined in the 3D model (Figure 1g). This stage helps to extract the positions of markers. The exported data of each marker is applied to the 3D model of each half (Figure 1h). The georeferenced point clouds of each half were in the same coordinate system. The exterior bodies of a rough surface were removed and only the rough surface remained. The height distance between the top and bottom rough surfaces was calculated (Figure 1i). To measure Joint Roughness Coefficient (JRC), three profiles in two perpendicular directions were drawn on the top and bottom rough surfaces (Figure 1j). The extracted profiles were digitized and JRC values were estimated (Figure 1k).

### 2.1. Sample Preparation

A rock sample of Kuru grey granite extracted from the Finnish bedrock was selected for this study. The sample was homogeneous with no visible pre-existing defects. A tensile throughgoing fracture was created with a manual plug and feather technique. Splitting was done by drilling holes at 15 cm spacing along the line of the planned fracture trace and hammering steadily the plug until the fracture propagated to the end of the block. Finally, the block was cut into 50 cm × 50 cm × 20 cm dimensions (Figure 1a).

### 2.2. Photography Procedure and Data Acquisition

A Structure-from-Motion (SfM) photogrammetric approach is used to scan the geometry of the sample. In this case, a set of overlapping images is captured from equally spaced locations around the sample at two angles. Shape and position of an object are determined by reconstructing bundles of rays. A combination of each camera location, each image point, and the corresponding perspective center defines the spatial direction of the ray to the corresponding object point. From the intersection of at least two corresponding (homologous), spatially separated image rays, an object point can be located in three dimensions [57]. Photogrammetric recording of rock fracture surfaces aims to automate detecting the rock fracture surface roughness and physical aperture without any manual sampling and costly laboratory experiments. Photogrammetry produces high-resolution and accurate three-dimensional (3D) models from two-dimensional (2D) images.

Since the 3D object is reconstructed from 2D images, the quality of the 3D models depends on image resolution and sharpness, camera sensor size and quality, position of the camera, and sufficient coverage of the captured subject. Finally, a resultant reconstructed 3D model can be affected by other elements such as light sources, texture of the object recorded, image processing, and reconstruction software. In this study, cameras of varying image resolution and sensor size were selected to test the influence on photogrammetric reconstruction. The data were collected using five different cameras. Canon EOS5DS R DSLR camera has the best sensor and high-resolution images of good quality, and previous research [50] successfully used this camera for measurement of JRC, which was confirmed by shear tests. Therefore, the Canon EOS 5DS R DSLR camera was considered as a benchmark for this research. and data acquired by other low-cost cameras was compared with Canon EOS 5DS R DSLR data. Raspberry Pi High Quality, as a customized camera, and GoPro Hero 8, as an action camera, were selected, as they have similar resolutions and sensor sizes. Two smartphones were chosen with high resolution (Xiaomi 10T Pro) and normal resolution (iPhone 12 Pro Max), also with different sensor sizes. To reduce the risk of the noise as much as possible, the lowest ISO values were selected. The specifications of each camera are presented in Table 1.

The images were taken in the largest resolution available and JPEG or RAW formats (see Table 1). The Depth of Field (DoF) was estimated based on the distance between the camera and the focus point on a portion of the sample once the settings and the location of the camera have been chosen.

Slabs were compactly matched to each other and each of them were fitted using dual ring 12-bit circular targets for photogrammetry (Figure 1a). For the aim of keeping the consistency with the global system and to scale the 3D model and align the two halves that are scanned separately, the circular reference points were placed around the upper and lower halves of all samples (see Figure 1a,b). A set of 6 dual-ring 12-bit markers was distributed on each side of the sample (in total 24 markers for the sample). To minimize the possibility of false matches due to repeated targets, each marker was unique, so that it can be automatically detected by the photogrammetric software. The markers were also used to test the accuracy of reconstruction by measuring the distances between each marker. A more detailed description of the measurement is given in Section 2.4.

Light is important to reduce the presence of noise in the image and shadows on the fracture surface. As the photogrammetry was performed indoors, it was necessary to provide enough illuminance on the rock surface. Eight portable LED lights were placed around the sample on a circle with 2 m radius from the sample center, as shown in Figure 2a, to provide the necessary luminous flux. The luminous flux was measured before the capture with a lux meter placed at nine different points (Figure 2b) of the slab’s surface. The amount of each point is tabulated in Table 2.

The camera was mounted on a tripod in a fixed position and tilted at 30° towards the sample placed on a rotating platform (Figure 3). The shooting distance was selected so that the photographed object fits onto the picture (Table 3). The slab was rotated in increments of 9°, with pictures taken at each rotation step. After each full rotation, three extra photos were taken to close a loop. This helps the software match the first and the final of the taken photos for the reconstruction of a 3D model. The full rotation produces 43 images. After each 360° rotation, the camera height and vertical angle were adjusted to 60°, and the process was repeated. In total, 86 photos were captured of the bottom and top halves of the sample. In addition, a total of 86 photos were taken of the sample when two pairs were matched together.

### 2.3. Data Processing

The reconstruction of the 3D models was done using the photogrammetric software RealityCapture 1.2 [58]. First, both the top and bottom samples are photographed together, then the bottom, and finally the top sample. Each assembly produces a total of 86 photographs. All photos from varying camera positions (3 × 86 = 258) were used to align the sample pair together. Next, the bottom and top images were aligned separately from 86 images. This process results in three alignment components: bottom, top, and bottom and top together.

Since the top and bottom halves of the sample were scanned separately to reconstruct the fracture surface, it was necessary to develop an approach for accurate positioning and matching of the top and bottom digital models in space so that it matched their original position when they were physically matched together. This enabled to calculate the physical aperture of the fracture. Moreover, the accuracy of the point cloud can be affected by the photogrammetric process. Therefore, the use of control points and measured distances between control points is also necessary to improve the accuracy of alignment. The markers on the sample surface were detected from the photos automatically in the software and added as control points on the images. The distances between markers were measured by a caliper with the accuracy of 0.1 mm and added as measured distances between control points. Next, the control point type was changed from the tie point to the ground control so that the xyz coordinates of each point can be specified. In the model of the whole sample, one control point was set as the origin of the coordinate system (0,0,0) and the coordinates of two other control points were set, from which two lie on the same plane with the origin (Figure 1g). Finally, the alignment was updated so that the aligned model’s position and scale is updated. Next, the calculated coordinates of all control points were extracted and imported into the bottom and top aligned models so that each model is positioned correctly in space with the same coordinate system. Next, the 3D dense clouds were calculated and colorized on the highest quality settings. Finally, reconstructed 3D point clouds were exported in the *.xyz format (Figure 1h).

Further processing of the point clouds was done in CloudCompare software version 2.10.2 [59], which is one of the best softwares to extract and calculate geometrical features of point clouds. The point clouds were cleaned by the elimination of the unnecessary points, such as background or noise. During this process, the models were cropped to eliminate the side surfaces of the samples, leaving only the fracture surface for further analysis.

The surface point density was measured with CloudCompare by using Compute Geometric Features operations. A sampling radius of 1 mm^2^ was selected to evaluate the number of neighboring points for the estimation of surface density of each point cloud. Finally, the point clouds were rasterized using the rasterize function in CloudCompare with a specified grid interval. The rasterization was done with the grid intervals of 0.1, 0.25, 0.5, and 1 mm for measuring the physical aperture and 0.25, 0.5, and 1 mm for estimating the JRC values.

The models obtained from other cameras were compared to the Canon EOS 5DS R DSLR model—the Canon DSLR. For example, the difference between the bottom surface of the Canon DSLR and smartphone point clouds was measured to show the differences in accuracy. In addition, the physical aperture and JRC values obtained from point clouds of each camera were compared to JRC values measured from the Canon DSLR models.

### 2.4. Estimation of Physical Aperture

Cloud Compare software [59] was used to calculate the physical aperture distribution from the point cloud data of the top and bottom halves (Figure 1h,i) using the cloud-to-cloud distance feature. The physical apertures of all 3D models of the cameras were measured after rasterizing the bottom and top surfaces with same X and Y coordinates in CloudCompare. The physical aperture was estimated by calculating the mean distance between the two-point clouds along the Z direction.

### 2.5. Estimation of JRC Values and JRC_error_

Each surface of the digital sample was intersected with three parallel lines in direction X and three lines in direction Y to extract data points for the measurement of JRC values (Figure 4). The measurement lines are equidistant. The 2D cross-sectional lines were therefore drawn on the original samples’ point clouds and their data were extracted, similarly, to Figure 4.

In this study, the root mean square (RMS) of local slope of the profile (*Z*_2_) (Equation (1)) was used to calculate the JRC [9].
(1)$$Z_{2}={\left[\frac{1}{N{\left(P\right)}^{2}}\sum_{i=1}^{N}{\left(z_{i+1}-z_{i}\right)}^{2}\right]}^{\frac{1}{2}},$$
where *N* signifies the number of intervals along each section, *P* is the point interval, and *z_i_* is the height of the asperities corresponding to the height local point. Varying uniform point interval with 0.25, 0.5, and 1 mm was used based on Equations (2)–(4) proposed by Yu and Vayssade [10]:JRC = 60.32(Z_2_) − 4.51 (Point interval: 0.25 mm),(2)
JRC = 61.79(Z_2_) − 3.47 (Point interval: 0.5 mm),(3)
JRC = 64.22(Z_2_) − 2.31 (Point interval: 1.0 mm).(4)

Each line was divided into five 10-cm sections and JRC was calculated for each section. The overall JRC value for each line was calculated as an average of all sections. The Canon DSLR data was considered as the data closest to reality [50]. The JRC values obtained from the data reconstructed from images captured with other low-cost cameras (JRC_camera_) were compared with the JRC values obtained from the Canon DSLR images (JRC_DSLR_) to estimate the JRC_error_ (Equation (5)):(5)$${\mathrm{JRC}}_{\mathrm{error}}=\frac{{\mathrm{JRC}}_{\mathrm{camera}}-{\mathrm{JRC}}_{\mathrm{DSLR}}}{{\mathrm{JRC}}_{\mathrm{DSLR}}}.$$

## 3. Results and Discussion

### 3.1. Constructed 3D Models

Figure 5 depicts the reconstructed 3D point clouds from the images captured with the Canon DSLR. The topography of the surfaces was extracted from the top and bottom halves as the height along Z-axis in the same coordinate system (Figure 5c,d). The difference between the height values of the two surfaces approximates the physical fracture aperture.

The difference between the measured distances between markers obtained with the caliper and the calculated distances from photogrammetric software for each camera are given in Table 4. The mean, standard deviation (SD), and root mean square error (RMSE) of those distances were calculated. As can be seen, the Canon DSLR (high-resolution camera) has the least error, and the Raspberry Pi HQ (low-resolution camera) has the highest error. However, the range of those errors is under 0.5 mm.

Table 5 shows the surface point density of the fracture surface obtained by each camera, ranging from 0.48 million to 12.6 million points. The Raspberry Pi HQ camera produced the lowest mean point density and Xiaomi 10T Pro smartphone the maximum. This is related to the image resolution of the cameras as shown in Figure 6, where the surface density is linearly proportional to the image resolution. The cameras arranged based on mean point density from highest to lowest are as follows: Xiaomi 10T Pro, Canon DSLR, iPhone 12 Pro Max, GoPro Hero 8, and Raspberry Pi HQ. Even though the surface density of points in the Canon DSLR model is not the highest, it is much smoother than others due to superior image quality of the Canon DSLR camera.

Figure 7 illustrates the bottom surface models for each camera. The blue color represents the minimum density of points in the middle of the reconstructed 3D surfaces, and it is a pattern that repeats in all models. This is related to the specifics of the rotary table method where the edges of the sample are always closest to the camera and the center is the furthest. As a result, the pixel size of the sample’s center is smaller compared to the edges, which results in lower point density of the center.

Table 6 shows the rasterized cloud-to-rasterized cloud distance between each surface obtained by the different cameras and the Canon DSLR, for the bottom and top surfaces with different grid intervals. The mean, standard deviation (SD), and root mean square error (RMSE) of those distances is calculated. It can be observed that the RMSEs increase with an increasing grid interval. If the mean differences are zero and the standard deviation values are close to each other, it means the point clouds are alike. Figure 8 demonstrates the computed RMSE of the rasterized cloud-to-rasterized cloud distance acquired by different devices. The different cameras show the good performances with low RMSEs. However, the lowest RMSE values are obtained from Xiaomi 10T Pro camera. It means the data obtained with this device is the closest to the Canon DSLR. The cameras can be ranked according to RMSE values of Figure 8 as follows: Xiaomi 10T Pro camera, iPhone 12 Pro Max, GoPro Hero 8, and Raspberry Pi HQ camera. iPhone 12 Pro Max and GoPro Hero 8 cameras show differences of RMSE values for the bottom and top surfaces. For Xiaomi 10T Pro and Raspberry Pi HQ cameras, the results are similar for both surfaces.

Figure 9 shows the rasterized cloud-to-rasterized cloud distance with 0.1 mm grid interval between each device and Canon DSLR data. GoPro Hero 8 shows good results (green area) at the corners, but differences appear in the middle (yellow area) of the fracture surface (Figure 9a). One explanation for such a difference can be the wide angle of the GoPro Hero 8 camera, which results in large image distortions. The Raspberry Pi HQ results shows large height differences that occur locally, so that the point cloud height is lower and higher than the Canon DSLR model. The Xiaomi 10T Pro point cloud is slightly elevated compared to the Canon DSLR point cloud, but it is negligible based on obtained RMSE, SD, and mean values. The GoPro Hero 8 and iPhone 12 Pro Max values of differences are less than 1 mm at the highest and lowest spots.

### 3.2. Calculated Physical Aperture

Figure 10 and Table 7 show the mean rasterized cloud-to-cloud distance between each surface (physical aperture) obtained by all cameras for the bottom and top surfaces for different rasterization grid intervals. By decreasing the interval grid values, the fracture aperture values obtained by different devices converged (Figure 10) so that the smallest interval values resulted in more precise estimation of the aperture for all cameras. A smaller interval means that more measurement between top and bottom surfaces are made to get the mean value of the approximated physical aperture. By increasing the interval values, the range of resulting values increases. Physical aperture acquired by high-resolution cameras such as Canon DSLR, iPhone 12 Pro Max, and Xiaomi 10T Pro smartphones are more scattered than low-resolution cameras such as Raspberry Pi HQ and GoPro Hero 8. One plausible cause for this result is the lower surface point density of the models reconstructed from the low-resolution cameras, resulting in smoother surfaces and not enough data points to be averaged. The Xiaomi 10T Pro trend of physical aperture estimations for varying grid interval changes more steeply than others. On the other hand, for GoPro Hero 8 and Raspberry Pi HQ, those changes are smaller. These results indicate that to get better results with different cameras, the small interval grid should be select during the rasterization of the models. Figure 11 shows the rasterized cloud-to-cloud distance for each camera with a grid interval of 0.5 mm.

### 3.3. Calculated JRC Values

The 3D models were further analyzed to extract the roughness of the rock fracture surfaces. Similar to aperture measurements, the JRC evaluation was done for three different rasterization grid intervals: 0.25, 0.5, and 1 mm. The models were sectioned by the 2D profiles (L_1_ to L_6_), as illustrated in Figure 4. Then, the points along the 2D profiles (L_1_ to L_6_) were extracted to calculate JRC according to Equations (1) to (4). The Canon DSLR model was chosen as a benchmark model due to a previous study by ref. [50]. It was shown that the JRC values obtained using Canon DSLR camera are similar to the JRC values calculated from shear tests.

As can be seen in Figure 12, by increasing the grid interval, the JRC values are increased, except the values obtained from the Xiaomi 10T Pro model, where the JRC increased slightly. In addition, the data points from the Xiaomi 10T Pro model result in the highest JRC values. The JRC calculated with the smallest interval values for iPhone 12 Pro Max, Raspberry Pi HQ, and GoPro Hero 8 are smaller than Canon DSLR. However, by increasing the grid interval, the calculated JRC of those cameras becomes closer to each other and to the Canon DSLR JRC values. It seems that the formulas used for estimating JRC values are sensitive to the dense data in the models rasterized with a small interval (0.25 mm). The calculated JRC values (5 to 7) with 1 mm interval for GoPro Hero 8, Raspberry Pi HQ and iPhone 12 Pro Max models are almost equal to the JRC values (5 to 7) estimated from the Canon DSLR model with 0.25 mm interval. Therefore, the resolution and optical quality of a camera should be considered to evaluate JRC values accurately. It means that not only grid interval value plays a key role in the estimation of JRC values, but also other parameters, such as image resolution, optical quality of the camera sensor, and quality of the extracted point cloud should be considered.

The Canon DSLR JRC values are almost in the same range for 0.25- and 0.5-mm intervals. However, by increasing the interval to 1 mm, the JRC value increased around 1 (6 to 7).

The mean value of JRC_error_ of all 3D models for varying gird intervals were calculated according to Equation (5) (Figure 13). The JRC_error_ decreases when the grid intervals values increase. The JRC values are higher with minimum error if the large interval values are selected. It seems that by choosing the small interval, an underestimation error occurs for low-resolution cameras. The high-resolution cameras, for instance Xiaomi 10T Pro and iPhone 12 Pro Max, have lower errors (−40% to +40% for 0.25 mm interval and almost between −20% and +20% for 1 mm interval) than low-resolution cameras (−60% to −80% for 0.25 mm interval and −20% to −40% for 1 mm interval).

Figure 14a,b, Figure 15a,b and Figure 16a,b show the profile L6 with 0.25, 0.5, and 1-mm intervals for the bottom and top surfaces obtained with different cameras. The horizontal line shows the X-axis direction, and the vertical line represents the Z-axis direction. The two surfaces of the fracture are in the same coordinate system and the height is measured from the same reference level. Noticeably, there are some visual differences among those profiles. To compare those in detail, Figure 14c,d, Figure 15c,d and Figure 16c,d are drawn to illustrate the deviations of heights in the Canon DSLR profiles used as a benchmark and other cameras for different intervals. The range of deviation fluctuates between −1 and 1 mm, except for interval 0.5 mm iPhone 12 Pro Max profiles ranging from −2 to 2 mm. By increasing the gird interval, the deviation between the profiles becomes smoother, most likely due to lower amount of extracted data for larger grid intervals.

## 4. Conclusions

A new photogrammetric method to determine physical aperture and roughness of a rock fracture was developed. In the new method, the sample pair is photographed using a circular table three times: the bottom sample, the top sample, and both samples in contact. Markers are used to align the top and bottom scans together to capture both the top and bottom fracture surfaces.

Low-cost cameras and smartphones can be used for the evaluation of physical aperture and JRC of rock fractures based on the accuracy of the aperture and the indicated JRC measurements. The cameras from best to lowest quality are Canon 5DS R DSLR, Xiaomi 10T Pro, iPhone 12 Pro Max, GoPro Hero 8, and Raspberry Pi HQ camera, in terms of the precision of distance estimation from the 3D models compared to the distances measured manually with a caliper.

A strong influence of the rasterization grid interval was observed. To rasterize 3D point clouds obtained by the cameras, four different grid intervals of with 0.1, 0.25, 0.5, and 1 mm were used. Reducing the rasterization grid interval for physical aperture measurement results in less scattered results, with the results converging at the 0.25 mm interval. The rasterization grid interval 0.1 mm shows the best results and is recommended.

For the 3D model derived JRC calculation, rasterization with a grid interval set to 0.5 mm gave the most reliable results. Measured JRC values are nearly identical to the results obtained by Canon DSLR. However, for low-resolution cameras, a rasterization grid interval of 1 mm resulted in the best result.

## Figures and Tables

**Figure 1 sensors-22-04165-f001:**
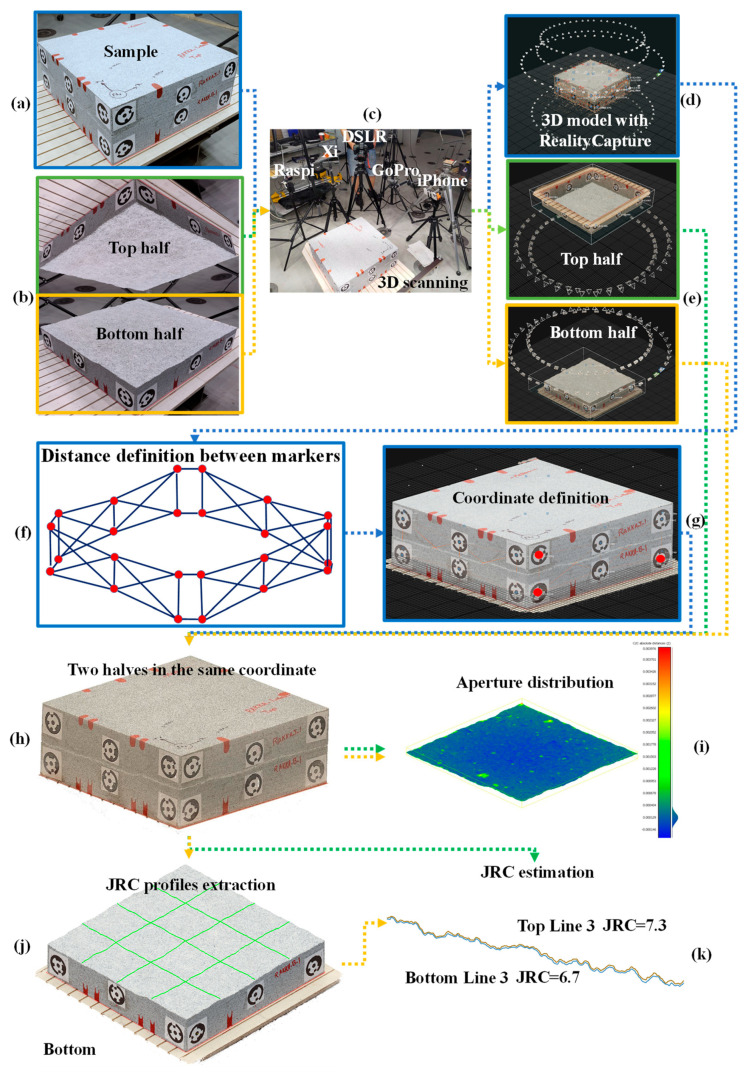
Workflow of the photogrammetric method to measure physical aperture: (**a**) Kuru grey granite sample with a throughgoing rough fracture, (**b**) top and half parts of the sample are scanned together and separately, (**c**) cameras used to capture the images of the sample, (**d**) photogrammetric processing and the calculated camera positions of the whole sample, (**e**) photogrammetric processing and the calculated camera positions of the two halves, (**f**) defined distances between markers on the whole sample, (**g**) defined coordinate system for the whole sample and extracting each marker position in the same coordinate system, (**h**) 3D point cloud of the sample with top and bottom halves matched in the same coordinate system with applying obtained markers’ positions for each half, (**i**) physical aperture distribution measured from the digital model, (**j**) profile extraction, and (**k**) Joint Roughness Coefficient (JRC) estimation.

**Figure 2 sensors-22-04165-f002:**
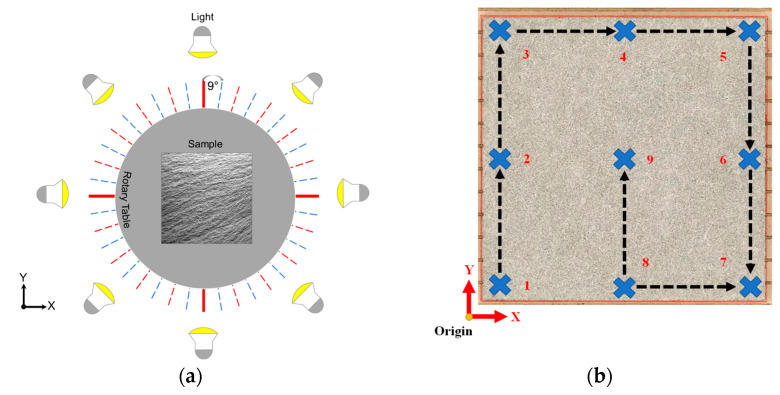
(**a**) LED light tripods around the slab and (**b**) the pattern of measuring the luminous flux on the sample surface as indicated by the numbers.

**Figure 3 sensors-22-04165-f003:**
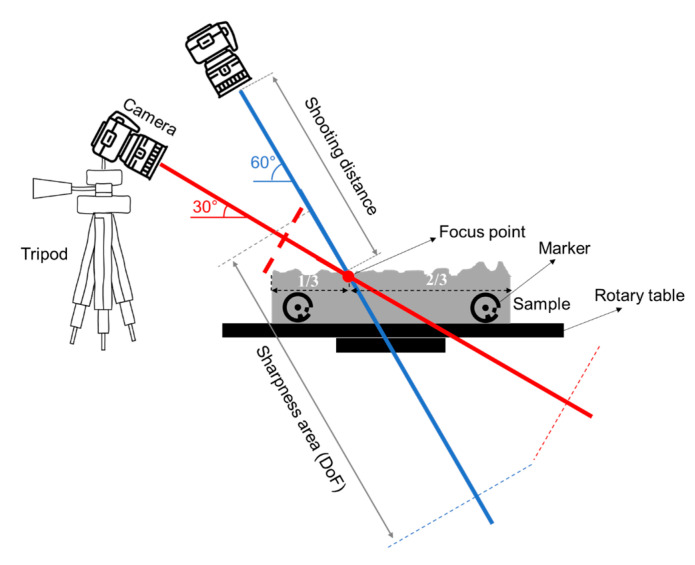
Sketch of the camera position in relation to the photographed sample.

**Figure 4 sensors-22-04165-f004:**
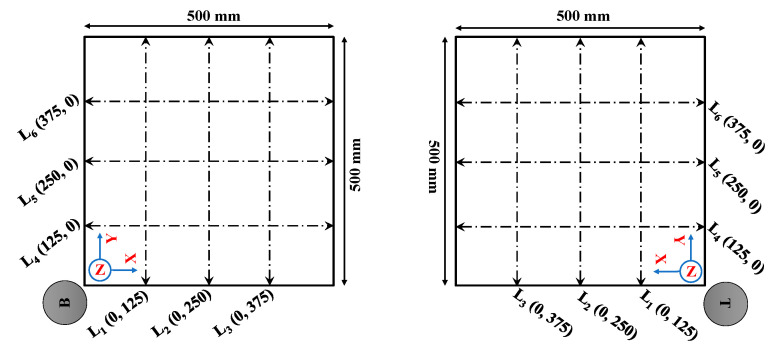
Visualization of the measurement lines for calculating JRC, B signifies the bottom and T is the top part of the sample.

**Figure 5 sensors-22-04165-f005:**
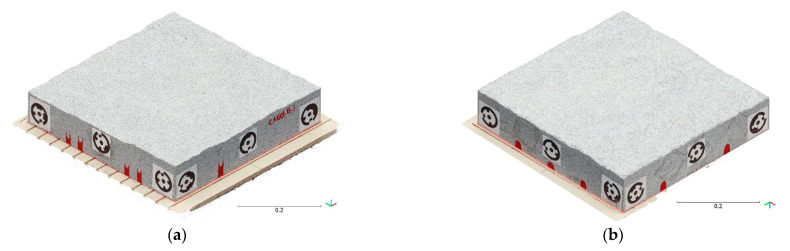
3D point clouds of the sample reconstructed from the images obtained using Canon DSLR camera, (**a**) the bottom half, (**b**) the top half, (**c**) topography of bottom surface (**d**) topography of top surface, and (**e**) the whole sample.

**Figure 6 sensors-22-04165-f006:**
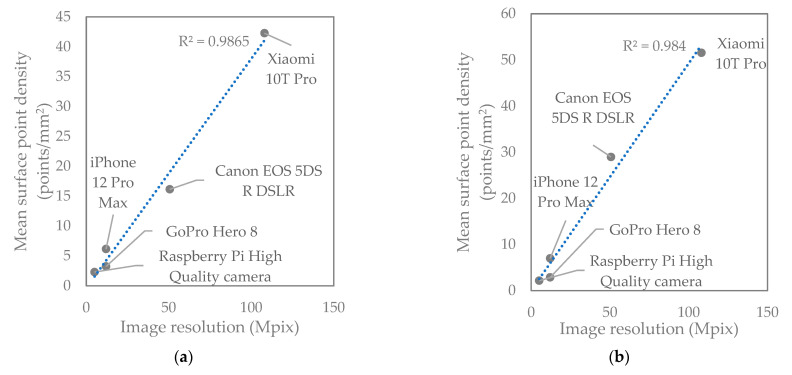
Mean surface point density of the top (**a**) and bottom (**b**) half of the sample as a function of camera resolution in megapixels.

**Figure 7 sensors-22-04165-f007:**
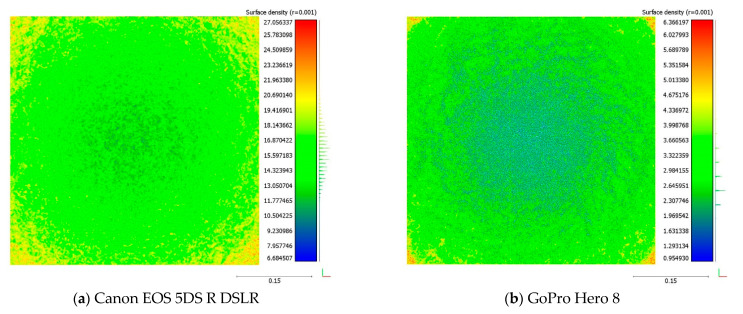
Surface density of the bottom surface in the 3D models.

**Figure 8 sensors-22-04165-f008:**
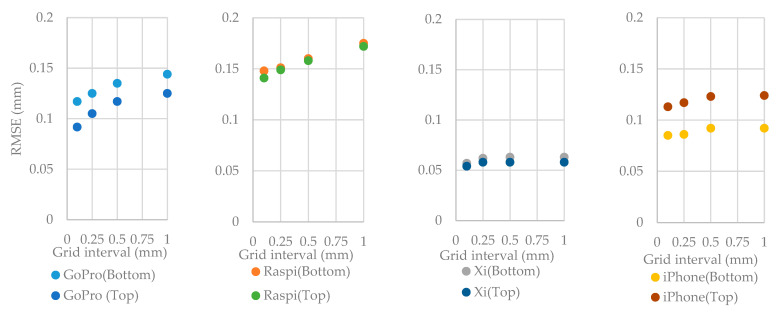
The root mean square error (RMSE) of the cloud-to-cloud distance between each low-cost camera model compared to Canon EOS 5DS R DSLR for the top and bottom samples with different rasterization grid interval.

**Figure 9 sensors-22-04165-f009:**
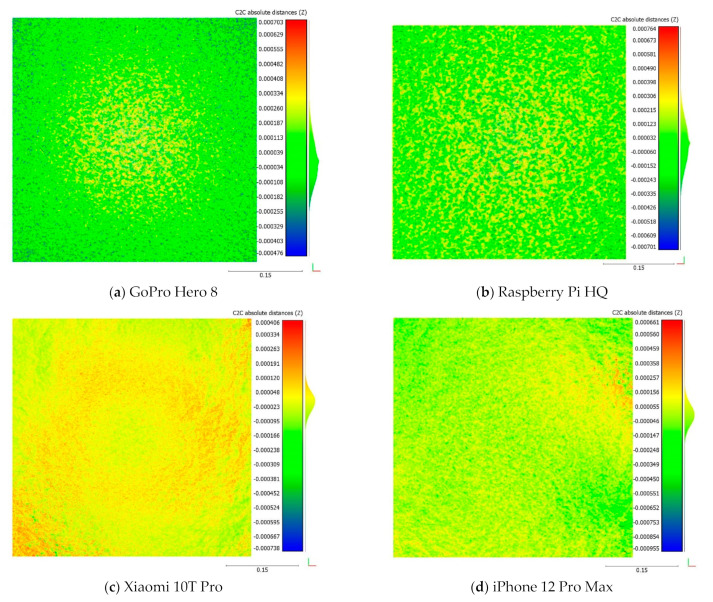
Comparison of cloud-to-cloud distance of low-cost camera point clouds to Canon EOS 5DS R DSLR point clouds of the bottom surfaces. The rasterization grid interval is 0.1 mm.

**Figure 10 sensors-22-04165-f010:**
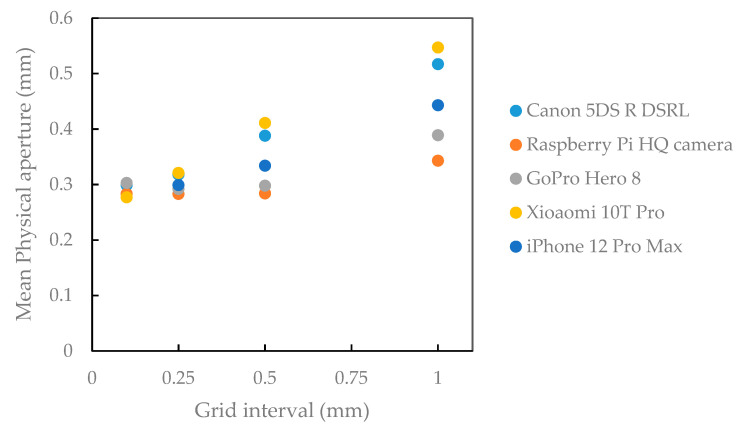
Comparison of estimated physical apertures obtained by all cameras for different rasterization grid intervals.

**Figure 11 sensors-22-04165-f011:**
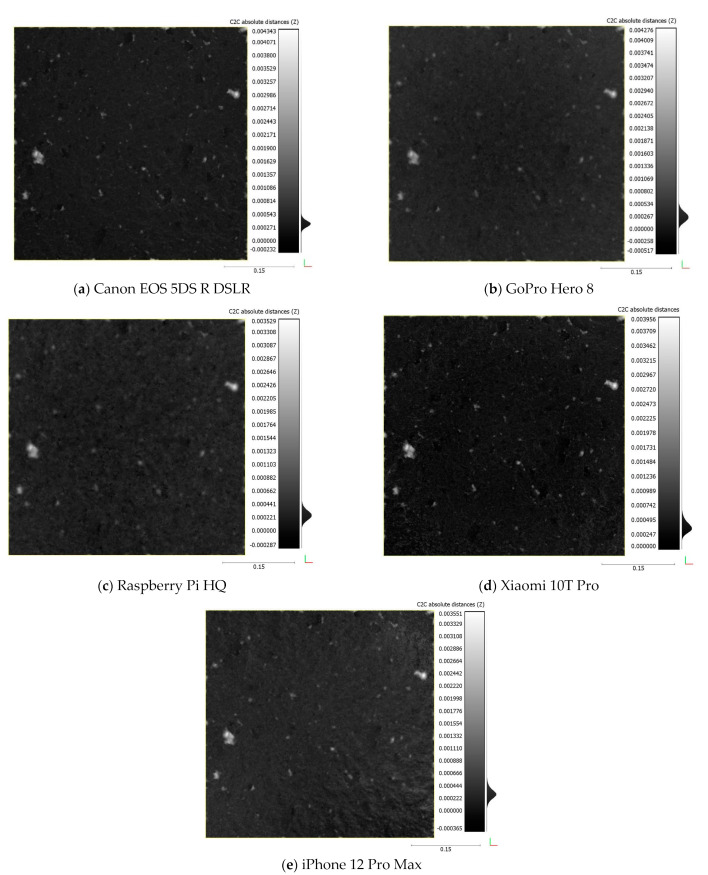
Physical apertures determined by cloud-to-cloud distance of bottom and top surfaces of each camera with a rasterization grid interval of 0.5 mm.

**Figure 12 sensors-22-04165-f012:**
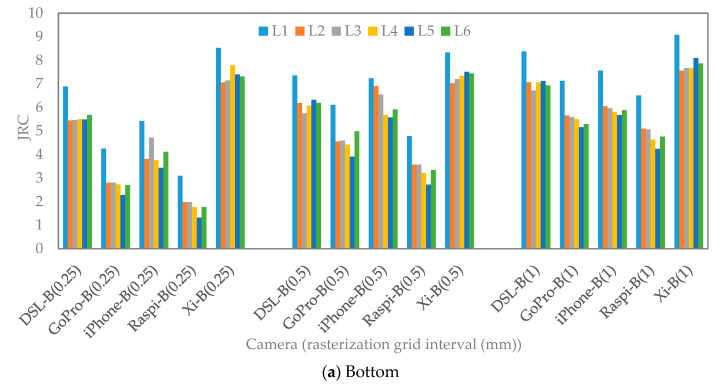
JRC values with different rasterization grid intervals obtained by different cameras alongside the 2D profiles (L_1_ to L_6_).

**Figure 13 sensors-22-04165-f013:**
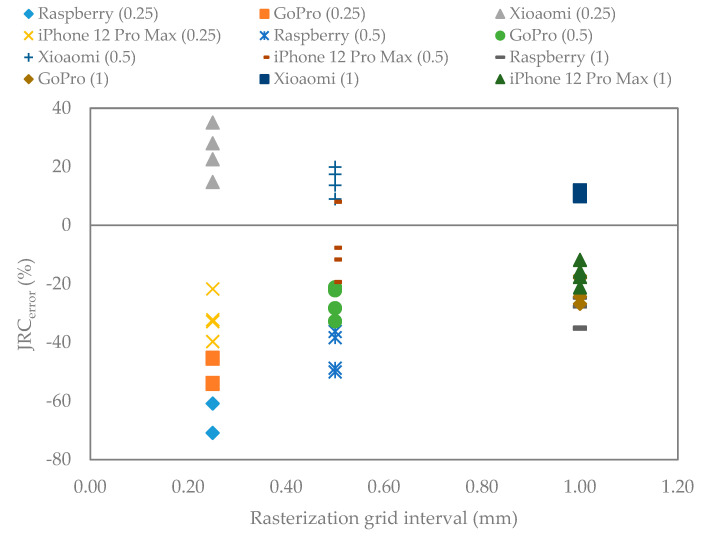
Mean value of the JRC_error_ extracted from the fracture 3D models rasterized with grid intervals set to 0.25, 0.5, and 1 mm.

**Figure 14 sensors-22-04165-f014:**
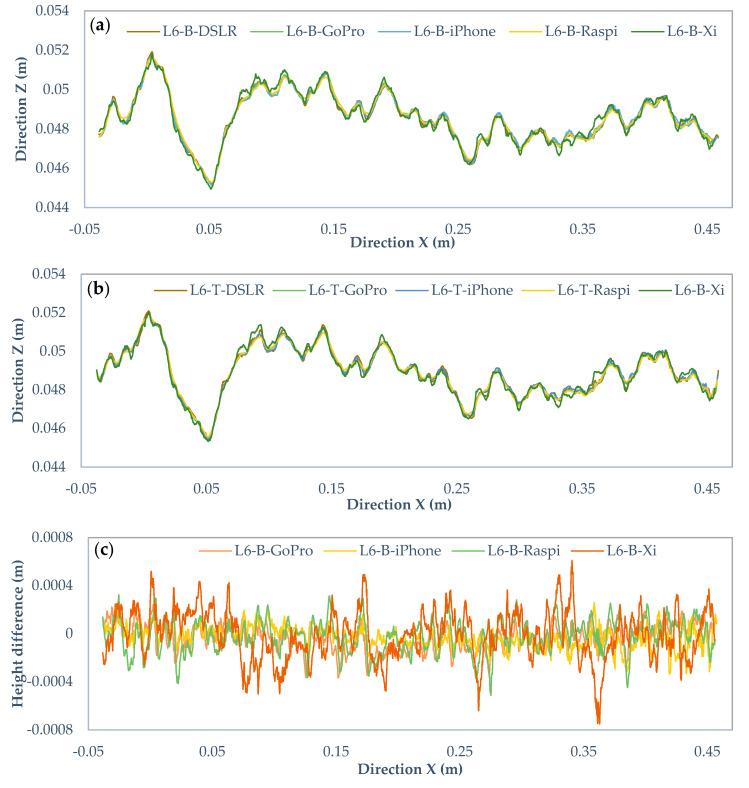
Comparison of the Canon EOS 5DS R DSLR and low-cost camera profiles with 0.25 mm rasterization grid interval. (**a**) Original profiles of bottom surface, (**b**) top surface, (**c**) height differences of bottom profiles, and (**d**) top profiles.

**Figure 15 sensors-22-04165-f015:**
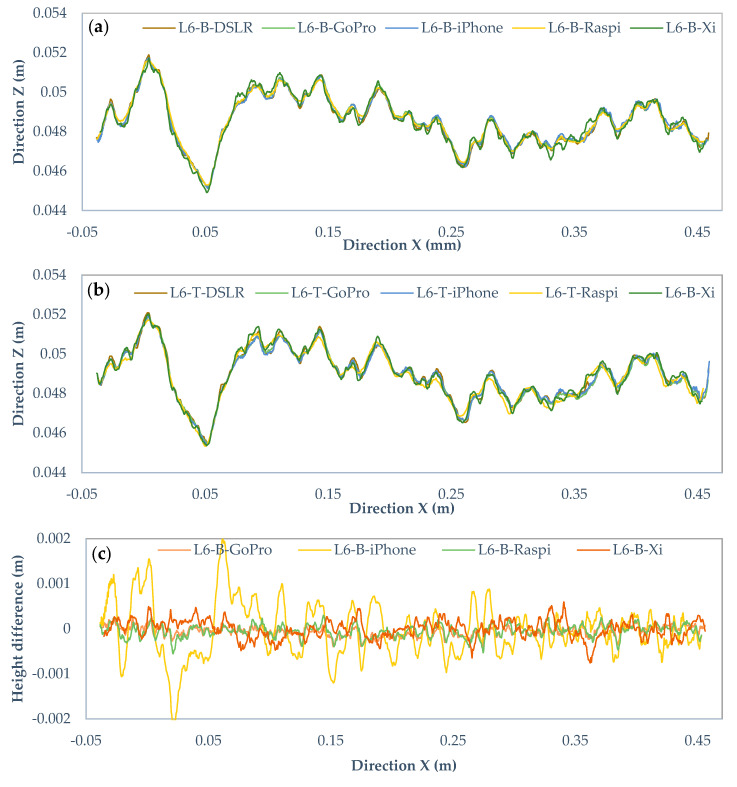
Comparison of the Canon EOS 5DS R DSLR and cameras profiles with 0.5 mm rasterization grid interval. (**a**) Original profiles of bottom surface, (**b**) top surface, (**c**) height differences of bottom profiles, and (**d**) top profiles.

**Figure 16 sensors-22-04165-f016:**
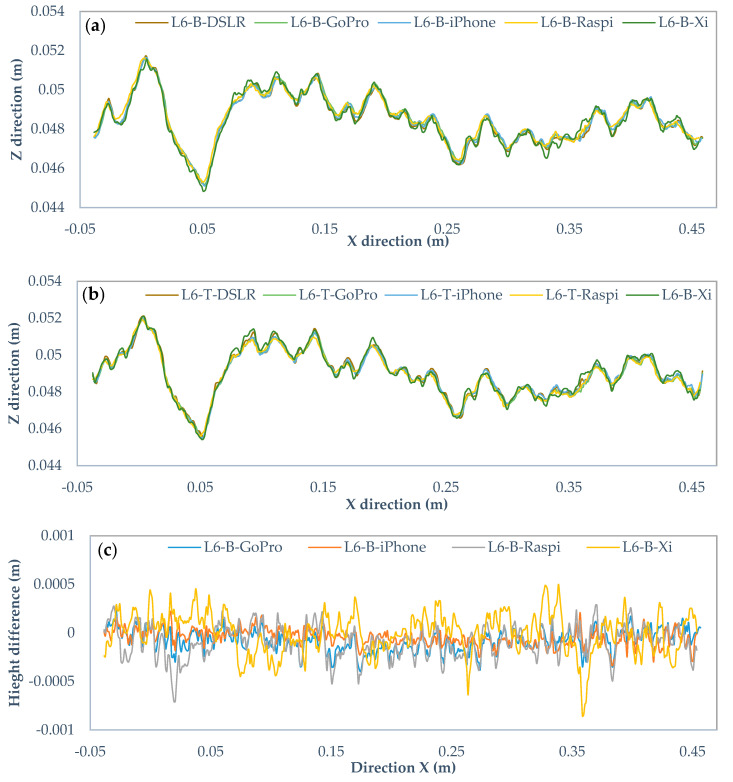
Comparison of the Canon EOS 5DS R DSLR and low-cost cameras profiles with 1 mm rasterization grid interval. (**a**) Original profiles of bottom surface, (**b**) top surface, (**c**) height differences of bottom profiles, and (**d**) top profiles.

**Table 1 sensors-22-04165-t001:** Camera specifications.

Item	Canon EOS 5DS R DSLR	Raspberry Pi High-Quality Camera	GoPro Hero 8	Xiaomi 10T Pro	iPhone 12 Pro Max
Max resolution (Mpix)	50.6	12.3	12	108	12
Sensor size (mm × mm)	36 × 24	6.287 × 4.712	6.16 × 4.62	9.6 × 7.2	6.85 × 5.14
Image resolution (pix)	8688 × 5792	2592 × 1944	4000 × 3000	12,032 × 9024	4032 × 3024
Pixel size (μm)	4.17	2.43	1.54	0.8	1.7
Lens	Canon EF 35 mm f/1.4 L II USM	Arducam 2.8–12 mm Varifocal	-	-	-
Lens focal length (mm)	35	3	3	7	5
35 mm equivalent	35	16.9	16.4	26	26
Format	JPEG + RAW	JPEG	JPEG	JPEG	DNG
ISO	100	100	100	50	32
F-Stop (aperture)	f/11	f/2	f/2.8	f/1.7	f/1.6

**Table 2 sensors-22-04165-t002:** Illuminance measurement results on the sample surface. Locations correspond to markings in Figure 2b.

Location	1	2	3	4	5	6	7	8	9
Illuminance (lux)	4010	4030	3920	4000	4040	4060	4000	4050	4130

**Table 3 sensors-22-04165-t003:** Shooting distances measured during data capture.

Camera	Shooting Distance (m)					
	Whole Sample		Bottom		Top	
	30°	60°	30°	60°	30°	60°
Canon EOS 5DS R DSLR	1.07	1.25	1.21	1.29	0.97	1.29
Raspberry Pi HQ camera	0.88	1.10	0.93	1.12	0.90	1.14
GoPro Hero 8	0.56	0.52	0.55	0.53	0.55	0.53
Xiaomi 10T Pro	0.72	1.10	0.75	0.97	0.79	0.98
iPhone 12 Pro Max	0.69	0.80	0.69	0.85	0.76	0.88

**Table 4 sensors-22-04165-t004:** The difference between measured and calculated marker spacing in 3D models produced using each camera.

Camera	Mean Control Distance Accuracy (mm)	SD (mm)	RMSE (mm)
Canon EOS 5DS R DSLR	−0.0016	0.230	0.228
Raspberry Pi HQ camera	0.3189	0.291	0.430
GoPro Hero 8	0.02163	0.232	0.230
Xiaomi 10T Pro	−0.0025	0.323	0.320
iPhone 12 Pro Max	0.0175	0.257	0.255

**Table 5 sensors-22-04165-t005:** Mean surface point density of the 3D point clouds of the fracture surface.

Specimen	Camera	Point Count	Mean Point Density (Point/mm^2^)	SD
Top	Canon EOS 5DS R DSLR	4,017,779	16.14	2.14
	Raspberry Pi HQ camera	503,843	2.299	0.34
	GoPro Hero 8	801,591	3.24	0.87
	Xiaomi 10T Pro	10,340,587	42.26	6.24
	iPhone 12 Pro Max	1,463,411	6.15	1.26
Bottom	Canon EOS 5DS R DSLR	7,186,385	28.97	4.45
	Raspberry Pi HQ camera	481,749	2.16	0.32
	GoPro Hero 8	734,769	2.88	0.6
	Xiaomi 10T Pro	12,558,026	51.56	8.45
	iPhone 12 Pro Max	1,674,754	6.99	1.50

**Table 6 sensors-22-04165-t006:** Comparison of cloud-to-cloud distance of low-cost camera point clouds in relation to the Canon EOS 5DS R DSLR point cloud.

Rasterization Grid Interval (mm)	Camera	Sample					
		Bottom			Top		
		Mean (mm)	SD (mm)	RMSE (mm)	Mean (mm)	SD (mm)	RMSE (mm)
0.1	GoPro Hero 8	0.004	0.117	0.117	−0.002	0.094	0.0917
	Raspberry Pi HQ	0.000	0.148	0.148	0.000	0.141	0.141
	Xiaomi 10T Pro	0.002	0.057	0.057	−0.004	0.054	0.054
	iPhone 12 Pro Max	0.001	0.085	0.085	−0.002	0.113	0.113
0.25	GoPro Hero 8	0.016	0.124	0.125	−0.013	0.105	0.105
	Raspberry Pi HQ	0.002	0.151	0.151	−0.009	0.149	0.149
	Xiaomi 10T Pro	−0.014	0.061	0.062	0.004	0.058	0.058
	iPhone 12 Pro Max	0.005	0.086	0.086	−0.010	0.116	0.117
0.5	GoPro Hero 8	0.046	0.127	0.135	−0.046	0.107	0.117
	Raspberry Pi HQ	0.034	0.156	0.160	−0.043	0.152	0.158
	Xiaomi 10T Pro	−0.022	0.065	0.063	0.007	0.057	0.058
	iPhone 12 Pro Max	0.023	0.089	0.092	−0.033	0.119	0.123
1	GoPro Hero 8	0.065	0.129	0.144	−0.064	0.107	0.125
	Raspberry Pi HQ	0.075	0.158	0.175	−0.078	0.153	0.172
	Xiaomi 10T Pro	−0.028	0.058	0.063	0.011	0.057	0.058
	iPhone 12 Pro Max	0.027	0.087	0.092	−0.040	0.117	0.124

**Table 7 sensors-22-04165-t007:** Measured physical aperture using different grid intervals.

Rasterization Grid Interval (mm)	Camera	Physical Aperture (Mean Value) (mm)	SD (mm)
0.1	Canon EOS 5DS R DSLR	0.299	0.184
	Raspberry Pi HQ camera	0.283	0.157
	GoPro Hero 8	0.303	0.202
	Xiaomi 10T Pro	0.277	0.174
	iPhone 12 Pro Max	0.297	0.179
0. 25	Canon EOS 5DS R DSLR	0.318	0.170
	Raspberry Pi HQ camera	0.283	0.148
	GoPro Hero 8	0.292	0.187
	Xiaomi 10T Pro	0.321	0.168
	iPhone 12 Pro Max	0.299	0.166
0.5	Canon EOS 5DS R DSLR	0.388	0.182
	Raspberry Pi HQ camera	0.284	0.152
	GoPro Hero 8	0.298	0.197
	Xiaomi 10T Pro	0.411	0.187
	iPhone 12 Pro Max	0.334	0.190
1	Canon EOS 5DS R DSLR	0.517	0.213
	Raspberry Pi HQ camera	0.343	0.168
	GoPro Hero 8	0.389	0.219
	Xiaomi 10T Pro	0.547	0.209
	iPhone 12 Pro Max	0.443	0.193

## Data Availability

Data is available on request.

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
