# Peer review of "Photogrammetric Method to Determine Physical Aperture and Roughness of a Rock Fracture"

_sensors, 2022, doi:10.3390/s22114165_

Round 1

Reviewer 1 Report

The authors proposed a novel method for determining the physical aperture and the roughness of a rock fracture. The manuscript is well-structured with a clear flow in explaining the background, methodology, results, and discussion. The reviewer has a few comments attached which may help to improve the quality of the paper.

1) The reviewer likes to see more discussion on SfM-based work in the field. Line 77 includes too many papers. It would be nice if the authors can add one more paragraph to distinguish your work from existing work in the field. Also, it might be wise to expand the discussion on the (2) contributions by explaining why low-cost cameras are important.

2) Followed by the comment above, regarding the second contribution, it is unclear to the reviewer how your study would be distinguished from existing work mentioned in Line 77. Since these works are also in SfM-based surface properties measurements.

3) A few questions for Fig. 1: why the rock sample has to be cut in half? is this required by the experimental protocol or something else? what is the rationale behind this workflow? are the point clouds for the top and bottom half created separately? what are the purposes of figures 1g and 1f? Currently, the reviewer has to go through individual sub-sections to look at the related discussion for each step in figure 1. It would be good if the authors can offer one paragraph upfront to explain the entire methodology of the method (i.e., figure 1). 

4) Line 131. The Canon camera could also contain errors. The accuracy of the point cloud could be affected by two factors. The first factor is driven by the image quality, test setup, lighting condition, and the number of images. The authors did address them in the manuscript. The second factor is caused by the drawback of photogrammetry as there are always errors that exist in point cloud reconstruction because of the limitation of SfM (or distortion in digital images).

5) Following the previous discussion in Item 4, would this factor lead to the error for the rock properties measurement? In remote sensing, folks use LiDAR and ground control points to address this concern. The GPS signal of ground control points will be collected to adjust the point coordinates in photogrammetry software. The authors may want to reflect on some of the discussions in the manuscript.

6) Section 2.4. How the aperture was calculated? Do the top and bottom models need to be aligned together? C2C distance of which planes shall be calculated?

7) Is there a way to obtain the physical aperture and roughness of the rock by other experimental methods? The reviewer was concerned that there could be some system errors related to photogrammetry (which is well-agreed in the field of remote sensing). 

8) How easily could the method be applied to other types of rock? How easily the method could be implemented in a more flexible setup (e.g., in the field with a hand-held camera, under normal lighting conditions). The authors might want to discuss these opportunities/challenges to broaden the application to the real world.

Author Response

Dear Reviewer 1, 

thank you for your comments. You can find the responses in the attachment. 

Best Regards  

Reviewer 2 Report

Dear author(s), please find some comments on the manuscript ‘Photogrammetric method to determine physical aperture and roughness of a rock fracture’, Manuscript ID: sensors-1703534:

  1. In the ‘Abstract’ it is difficult to find one or, respectively, the most crucial findings from the whole text. Even is interesting and contains all of the (required) relevant information, is not directed so, simultaneously, makes the reader confused. Please try to emphasize the most important studies and proposals instead of a detailed description.
  2. The “introduction’ section is appropriate, however, a more critical review would be preferred in its current form, both the novelty and motivation are not indicated from the (critical) analysis of the current stage of knowledge and already published (cited) items. Even interesting, this section must be improved (fulfilled).
  3. As in the previous comment, the proposal introduced in lines 89-97 (‘In this study, an attempt has been made to provide a methodology to characterize physical aperture using photogrammetry as a non-destructive method. Besides, a comparison study was conducted to identify the effect of using different photography devices with different resolutions and costs on characterizing geometrical properties of natural rock fractures. The aperture is calculated using the cloud-to-cloud distance of the top and bottom models and the surface roughness is calculated using the Z2 approach [9] and JRC formulas proposed by Yu and Vayssade [10]. The effects of the photogrammetric reconstruction and model rasterization on the measured aperture and roughness are assessed. Finally, some of the benefits and drawbacks of employing this method are highlighted.’) is not received from the previous part (lines 28-88) of this section. Please try to emphasize lack of the knowledge presented in this manuscript but missed in the literature.
  4. More details of noise mentioned in line 166 should be provided. Even the LED light tripods could provide a suitable light, some risk of noise data should be introduced, if exist.
  5. There are many shortcuts and abbreviations so, respectively, additional sections (e.g. ‘Abbreviations’) would be required that in its current form, makes the regular reader confused and, unfortunately, often lost.
  6. The ‘Conclusion’ section is interesting, nevertheless, should be divided into numbered gaps to highlight both the novelty and studies performed.
  7. According to the references, only 14 from 54 are from the last 5-6 years, and many of the rest are more than 10 years ago. From that matter, I ask if the topic can be classified as up-to-date?
  8. Continuing with references, there is only one paper found from the Sensors journal but, correspondingly, most of the cited items are from the 3-4 journals. From that point of view, does the manuscript better fit the scope of the Sensors journal or those (often cited) journal(s)? I feel strongly confused (!) so, in my opinion, this manuscript better fits for the journal(s) considering materials properties than the measurement science, which Sensors presents.

Moreover, some additional, editorial issues must be mentioned:

  1. In line 284 there is a double space.
  2. There are (unrequired) gaps in some cases, like between lines 339 and 340.
  3. The ‘Reference’ section should be unified according to the journal template requirements.

Concluding, the manuscript is interesting, however, in its current form, is not suitable for publication in a quality journal as the Sensors is but, respectively, can be further processed, if a major revision is allowed.

Author Response

Dear Reviewer 2,

thanks for the comments. You can find the responses in the attachment.

Best Regards 

Round 2

Reviewer 1 Report

I recommend this paper to the journal.

Author Response

Thank you very much.

Reviewer 2 Report

Dear author(s), a manuscript titled ‘Photogrammetric method to determine physical aperture and roughness of a rock fracture’, Manuscript ID: sensors-1703534, has been improved in a required, suitable manner, therefore can be further processed by the Sensors journal.

However, before the final decision (acceptation, if allowed), want to ask you for the last modification that, in my opinion, the ‘Conclusions’ section is too weak, and has some general responses. Moreover, it looks too long but does not convince the regular reader. This section does not correspond to the whole manuscript, which is much better than the sentences presented in this, last section. Therefore, try to emphasize novelty more strongly. Moreover, conclusions should be divided into separate, numbered gaps, as mentioned in the previous round of review but, unfortunately, not taken into consideration by the author(s).

Taking the above suggestion, the manuscript can be accepted for publication in a quality journal as the Sensors is.

Further, thank you for your responses that, in their current form, were addressed properly and make the manuscript more suitable for publication in the Sensors journal.

Author Response

As the respected reviewer 2 asked; the conclusion is modified according to that format. It is shortened and divided into numbers. 

4. Conclusions
1. A new photogrammetric method to determine physical aperture and roughness of a rock fracture was developed. In the new method, the sample pair is photographed using a circular table three times: the bottom sample, the top sample, and both samples in con-tact. Markers are used to align the top and bottom scans together to capture both the top and bottom fracture surfaces.
2. Low-cost cameras and smartphones can be used for the evaluation of physical ap-erture and JRC of rock fractures based on the accuracy of the aperture and JRC measure-ments indicate. The cameras from best to lowest quality are Canon 5DS R DSLR, Xiaomi 10T Pro, iPhone 12 Pro Max, GoPro Hero 8, and Raspberry Pi HQ camera, in terms of the precision of distance estimation from the 3D models compared to the distances measured manually with a caliper.
3. A strong influence of the rasterization grid interval was observed. To rasterize 3D point clouds obtained by the cameras, four different grid intervals of with 0.1, 0.25, 0.5, and 1 mm were used. Reducing rasterization grid interval for physical aperture meas-urement, results in less scattered results with the results converging at the 0.25 mm inter-val. The rasterization grid interval 0.1 mm shows the best results and is recommended.
4. For the 3D model derived JRC calculation, the rasterization with grid interval set to 0.5 mm gave the most reliable results. Measured JRC values are nearly identical to the re-sults obtained by Canon DSLR. However, for low-resolution cameras, the rasterization grid interval 1 mm resulted in the best result.

Thanks for your comments that helped to improve our paper.